# Do children interpret informants' confidence as person-specific or situational?

**Aimie-Lee Juteau**[ID][1]*, **Yasmeen A. Ibrahim**[2], **Sara-Emilie McIntee**[ID][1◉], **Rose Varin**[3◉], **Patricia E. Brosseau-Liard**[1]

1 School of Psychology, University of Ottawa, Ottawa, Ontario, Canada, 2 Department of Psychology and Neuroscience, Dalhousie University, Halifax, Nova Scotia Canada, 3 Psychology Department, University of Montreal, Montreal, Québec, Canada

◉ These authors contributed equally to this work.
* ajute095@uottawa.ca

## Abstract

Children prefer to learn from confident rather than hesitant informants. However, it is unclear how children interpret confidence cues: these could be construed as strictly *situational* indicators of an informant's current certainty about the information they are conveying, or alternatively as *person-specific* indicators of how "knowledgeable" someone is across situations. In three studies, 4- and 5-year-olds (Experiment 1: $N = 51$, Experiment 3: $N = 41$) and 2- and 3-year-olds (Experiment 2: $N = 80$) saw informants differing in confidence. Each informant's confidence cues either remained constant throughout the experiment, changed between the history and test phases, or were present during the history but not test phase. Results suggest that 4- and 5-year-olds primarily treat confidence cues as situational, whereas there is uncertainty around younger preschoolers' interpretation due to low performance.

**Data Availability Statement:** The data that support the findings of this study is available in the Open Science Framework public repository and can be found here https://osf.io/9a4j2/.

## 1. Introduction

Knowledge acquisition is a continuous process that begins at a very early age. It is not always possible for children to obtain information first-hand. Hence, they learn about various topics from people around them [1], often through observation, imitation, and trust in information that is given to them. Social learning can also be a selective process that allows children to be skeptical when assessing the credibility of the informants providing them with information (see Clément [2] for a review). Indeed, people are fallible and can intentionally or inadvertently mislead children. Research has widely corroborated that children engage in *selective social learning*, that is, the inclination to learn from or prefer some sources of information over others [3–5].

Many cues can help children decide whether or not to learn from a given individual. For instance, preschool- and young school-age children appear to prefer learning from those who are nicer (3- to-5-year-olds) [6], more attractive (4- to-5year-olds) [7], and who share their language or accent (3- to-6-year-olds) [8, 9]. Some of these cues may seem to have little relation to the likelihood that individuals provide adequate information; yet, there are numerous cues children can use that can be considered *epistemic* indicators, or likely indicators of how

**Funding:** This work was supported by the Natural Science and Engineering Research Council of Canada under the Discovery Grant [grant 2015-05775](https://www.nserc-crsng.gc.ca/professors-professeurs/grants-subs/dgigp-psigp_eng.asp) awarded to the last author (PBL). The funders had no role in study design, data collection and analysis, decision to publish, or preparation of the manuscript.

**Competing interests:** The authors have declared that no competing interests exist.

knowledgeable the individual providing information is. For example, many studies have shown that, before the age of two, young children can detect and use individuals' past reliability [10–13], expertise [14], and age [15], all of which likely correlate with the individual's knowledge, to decide whether to learn from that individual or not.

The cue of particular interest in the present research is the expressed *confidence* of an individual. When one is knowledgeable, one frequently expresses certainty through non-verbal indicators, such as nodding, an upright posture, and a rapid, assured tone of voice, accompanied by verbal confidence markers such as "I know" or "for sure"; conversely, when one is ignorant or uncertain, this is commonly expressed by verbal hedging such as "I guess" or "maybe", combined with paralinguistic and non-verbal markers such as a hesitant tone, shrugging, and a puzzled facial expression [16]. By 24 months, toddlers are more likely to imitate the actions of an individual expressing non-verbal confidence cues over a counterpart portraying uncertainty [17, 18]. Preschoolers aged 3 to 5 years old also prefer to learn from those who express their certainty both verbally and non-verbally over those who express hesitation or uncertainty [16, 19–21]. Of course, confidence is far from a perfect knowledge cue: whether intentionally or not, an individual could be poorly *calibrated*, that is, showing confidence cues that are disproportional to their true knowledge [22]. There is evidence that older preschoolers, aged 5, possess some understanding that confidence is not as reliable a credibility cue as, for instance, an individual's history of accuracy when the two conflict directly [23]; however, this remains challenging to them when the relation between an individual's accuracy and confidence is complex [24].

Though multiple studies have shown that children do attend to confidence cues and prefer, all else being equal, to learn from confident over hesitant individuals, it is unclear exactly how they interpret or understand confidence. One specific aspect of their interpretation that remains unclear is how much children believe an individual's confidence cues indicate something durable about that person's trustworthiness. When describing the different epistemic cues that one could use to evaluate an individual's credibility, Miller [25] drew a distinction between *situational* and *individual* cues to knowledge (to avoid confusion with other uses of the term "individual", we will refer to the latter as *person-specific* instead, as per Brosseau-Liard & Birch [26]).

Originally, most research on children's attributions of knowledge studied state-specific or *situational* indicators of knowledge—cues in a specific situation that indicate whether or not someone has the knowledge pertinent to that situation, regardless of how generally "knowledgeable" they might otherwise be. Many of these situational cues pertain to information accessed through perceptual or other means. For instance, a person looking inside a box has knowledge of its contents, whereas another person who has not looked inside that box does not know what is inside. Traditional false-belief tasks [27] can be seen as evaluating children's understanding of others' situational knowledge or lack thereof [28, 29]. Research has generally demonstrated that preschoolers have a nascent but imperfect understanding of the relation between information access through various perceptual means and knowledge [30, 31]. They can also understand more indirect situational knowledge indicators: For example, they understand that familiarity with an item makes one knowledgeable about it, without necessarily needing to identify the exact perceptual means of knowledge acquisition [32].

Whereas *situational cues* are only informative of someone's knowledge in a particular situation and can be seen as independent of an individual's personal attributes, *person-specific cues* are informative about an individual's overall knowledge across situations [33]. Indeed, some people know more than others, and observable attributes can help children figure out whether a given individual is more or less likely to be knowledgeable or ignorant. For instance, an individual's age can be a proxy for knowledge: children prefer to learn from adults over children

[34, 35]. A person's past accuracy, domain of expertise, and social status can all be considered person-specific cues to that person's knowledge, and all can be used by children in the pre-school years at least in some situations (see Mills [5], for a review).

What about confidence? Although research has demonstrated that children attend to cues of confidence, it is unclear whether they treat them as situational or person-specific. Unlike many other cues, confidence could conceivably be interpreted both ways. On one hand, children could interpret confidence or hesitance cues as indicators of how a person feels *right now* about the knowledge they are *currently* sharing, and not transfer this information to any other situation involving the same person sharing other information. Alternatively, children could make broader attributions based on the same cues: they could assume, for instance, that some-one who expresses themselves confidently is a person who is generally smart, knowledgeable or in possession of a high social status granted by expertise, while one who is hesitant could be perceived as being a more generally ignorant person. In other words, expressions of confi-dence in one situation could be interpreted as indicative of someone's knowledge in other situations.

Importantly, the way children perceive confidence could lead to significant mistakes should they not understand to what extent confidence is an "imperfect" cue to knowledge. Indeed, although older children can understand the concept of *calibration* (i.e., refers to how propor-tional confidence is compared to knowledge [24, 36] weighing different credibility cues is diffi-cult for younger children, and interpreting confidence cues as person-specific could result in misinformed learning decisions. Simply picture how confident some public figures appear about their own knowledge even when they are speaking on topics outside their field of exper-tise, and how even otherwise savvy adults can be misled by misplaced confidence. The same could apply to young children in regard to their teachers, instructors, parents or peers. If chil-dren perceive confidence as indicative of an individual being generally knowledgeable across situations, this could, in some circumstances, lead to misinformed learning decisions (e.g., believing and learning from someone only because they previously expressed confidence while they might not actually know what they are talking about). On the contrary, if children per-ceive confidence as situational, they may be a bit less vulnerable to this specific type of misin-formation: They would instead perhaps rely on other cues, such as expertise, to decide whether to believe someone or not.

To date, however, most research investigating children's use of confidence cues has kept individuals' confidence constant, therefore making it difficult to tell whether children consider both individuals' past and current confidence cues when making judgments of trustworthi-ness. To our knowledge, the only exception is a series of studies by Moore and colleagues, which examined children's decision-making based on puppets' verbal (lexical and prosody cues) cues of certainty [37, 38]. In these studies, participants had to choose the box in which they believed there was candy based on statements from two puppets and with no opportunity to receive feedback in between trials. Statements from puppets implied contrasting cues of cer-tainty and the puppet making the more certain statement varied across the twelve trials of the test session. Moore and colleagues found that four- and five-year-olds most often made their decision based on the puppet that used stronger lexical cues of certainty in the immediate trial [37–39]. For instance, in one study [39], four- and five-year-olds favored information from whichever puppet implied certainty through a statement using "know" instead of "think, and did not appear to do so more or less in the first versus the second half of the study. However, these studies were not specifically designed to test the impact of prior or changing confidence on children's trust. Furthermore, guessing the transient location of a piece of candy involves a highly situational focus: some past research has found that children show less generalization of

**Table 1. Summary of conditions and test phases across experiments.**

| Experiments | Tasks at test phase | Conditions | Confidence in History Phase | Confidence at Test Phase |
|---|---|---|---|---|
| 1 | Word learning task | Consistent confidence | • Informant 1 is confident<br>• Informant 2 is hesitant | • Informant 1 remains confident<br>• Informant 2 remains hesitant |
| | | Inconsistent confidence | • Informant 1 is confident<br>• Informant 2 is hesitant | • Informant 1 switches to being hesitant<br>• Informant 2 switches to being confident |
| 2 | Part 1: Word learning | Consistent confidence | • Informant 1 is confident<br>• Informant 2 is hesitant | • Informant 1 remains confident<br>• Informant 2 remains hesitant |
| | | Inconsistent confidence | • Informant 1 is confident<br>• Informant 2 is hesitant | • Informant 1 switches to being hesitant<br>• Informant 2 switches to being confident |
| | Part 2: Imitation | Consistent confidence: confident | • Familiar Objects phase: Informant 3 is confident | • Novel Objects phase<br>• Informant 3 remains confident |
| | | Consistent confidence: hesitant | • Familiar Objects phase: Informant 3 is hesitant | • Novel Objects phase<br>• Informant 3 remains hesitant |
| | | Inconsistent confidence: confident to hesitant | • Familiar Objects phase: Informant 3 is confident | • Novel Objects phase<br>• Informant 3 switches to being hesitant |
| | | Inconsistent confidence: hesitant to confident | • Familiar Objects phase: Informant 3 is hesitant | • Novel Objects phase<br>• Informant 3 switches to being confident |
| 3 | Part 1: Word learning<br><br>Part 2: Attributions | History-only condition | • Informant 1 is confident<br>• Informant 2 is hesitant | Neither informant is confident nor hesitant (all information is presented via text) |

cues involving episodic information (e.g., where a piece of candy is hidden right now) compared to semantic information such as object labels and functions [26, 40–42].

In the current research, we examined children's use of individuals' changing confidence cues when learning semantic information by comparing conditions where children learn from individuals with stable versus changing confidence during an experiment. Children witnessed a history phase where two individuals provided information, one confidently and the other hesitantly. At test, the same individuals either continued showing the same confidence cues (Experiments 1 and 2), switched from confident to hesitant and vice versa (Experiments 1 and 2), or demonstrated neither confidence nor hesitance (Experiment 3). Table 1 presents a summary of conditions and phases across experiments (note that the identities of "Informant 1" and "Informant 2" were counterbalanced in all conditions where two informants were present). Experiments 1 and 2 were conducted concurrently but with two different age groups. Results from these experiments led to a further exploration of older children's performance in Experiment 3. If children perceive confidence as a strictly situational knowledge cue, they should *only* use the informant's level of confidence expressed during the test phase to decide which informant to learn from as it reflects the current knowledge of the informant for that specific situation. Therefore, prior confidence should have no impact whatsoever on their learning. On the other hand, if children interpret confidence cues as indicative of a durable person-specific characteristic, we would expect the informant's confidence level during the history phase to continue affecting children's responses at test. Data collected in the three experiments presented below are available at the following link (https://osf.io/9a4j2/?view_only=dc7e670c756c41ceb8e84735faf397ad) [43].

## 2. Experiment 1

In the first experiment, we study the preferential trust that older preschoolers (ages 4 and 5) show towards informants whose confidence cues either remain constant or vary. We devised a

new manipulation containing two experimental conditions: *consistent* and *inconsistent* confidence. The consistent confidence condition presented children with two individuals who provided semantic information, one while being consistently confident and the other consistently hesitant. This condition simply aimed to replicate prior research demonstrating that young children prefer to learn from a confident rather than a hesitant individual. In the inconsistent confidence condition, we presented children with informants who expressed changing levels of confidence; i.e., one informant initially expressed confidence but became hesitant during test trials, whereas the second informant started out hesitant and became confident. Children's performance when they must choose between a currently confident but previously hesitant informant and a currently hesitant but previously confident one can indicate whether they treat confidence as a situational or person-specific indicator. This experiment was preregistered following partial data collection (26 of the 51 participants were tested prior to preregistration).

## 2.1 Method

**2.1.1 Participants.** We preregistered a sample of 48 participants (link to preregistration: https://doi.org/10.17605/OSF.IO/MX3DW). Because we scheduled more participants than needed in order to account for cancellations and eliminations, the final sample included 51 typically-developing 4- and 5-year-olds (48–71 months, $M_{age}$ = 61 months; 25 boys and 26 girls). Participants were recruited via an in-lab participant database between June 30, 2016 to July 4, 2019. Children were predominately White (nine reported mixed ethnicity, two Black and one South Asian; 12 did not report race/ethnicity) and came in majority from families of average to above-average income (four did not report family income). All children spoke English, the language of the study. Data from six additional participants were excluded due technical difficulties (1), failure to answer one or more test trials (2), having participated twice (1), poor language comprehension (1) and falling outside the target age range (1).

**2.1.2 Materials and procedure.** Before participating in the study with their child, parents provided their written consent by completing a consent form. Children's verbal assent was also obtained before starting the testing sessions. Children were tested individually in a quiet room on a university campus. Each child sat in front of a laptop or tablet computer screen with the experimenter sitting beside them. When parents were present in the room, they were asked not to interact with the participant during the study. Experimental sessions were video recorded with a camera placed in a way that allowed for an unobstructed view of the experimental setting. The recorded videos were later used to verify the participants' answers. This study was part of a larger experiment including additional conditions not reported or described here.

*Introduction phase.* Children were introduced to a game where they were told they would have to name objects pictured on cards that were placed in front of them. Children were told they would be rewarded with stickers every time they named an object correctly. To familiarize children with the fact that they would see unknown objects, we presented both highly familiar pictures (e.g., an apple) and unfamiliar ones (e.g., a coffee roaster) and asked them to name the pictures, providing feedback each time. Subsequently, children were introduced to a puppet who they were told would decide on the administration of stickers for subsequent trials. They were informed that the puppet was very nice and would reward them with stickers regardless of their answers.

*History phase.* Children were told that, for subsequent trials, they could receive help from two adult female informants presented on video (hereafter labelled B and S). They watched two brief videos introducing each informant waving, and then watched a series of videos in

which B and S named five familiar objects (e.g., dress, shoes, desk) while holding pictures that weren't visible to the child (so that the child could not tell whether the informants were accurate or not). One informant was confident (e.g., saying: "Oh I know, that's a dress!" with a declarative tone, while holding a picture not visible to the child, raising the index finger and having a satisfied facial expression) and the other was hesitant (e.g., saying: "I. . . guess that's a dress?" with an upward inflection, while also holding a picture not visible to the child, shrugging shoulders and having a puzzled facial expression). The informants were always presented in the same order, B then S, but which informant served as the confident informant was counterbalanced to account for order effects.

*Test phase.* After watching the history videos, five *endorse* test trials were conducted. In each trial, a picture of an unfamiliar object was placed in front of the participants. The experimenter then played a video of each informant proposing a different made-up label for the same novel object, one confidently, the other hesitantly. Confidence and hesitance were expressed using the same cues portrayed during the history phase. For instance, B would refer to an object saying "Oh I know, that's a *heem*!", whereas S would say "I. . . guess that's a *bloof*?". Subsequently, the experimenter asked children to label the unfamiliar object using one of the novel labels provided by the informants. The same procedure was repeated in all five trials.

The experiment had a between-subjects design and participants were randomly assigned to one of two conditions: consistent confidence or inconsistent confidence. In the consistent confidence condition, the informant who was confident during the history phase remained confident during the test phase and so forth with the hesitant informant. In the second condition, inconsistent confidence, the informant who was confident during the history phase became hesitant during the test phase and vice-versa.

## 2.2 Results and discussion

As preregistered, we calculated the total number of trials (out of 5) on which participants selected the same answer as the *currently* confident informant, regardless of their past confidence. The mean was compared to chance (2.5) with a directional one-sample t-test, revealing that children overall did, as expected, prefer to learn from the currently confident informant over the currently hesitant one, $M = 3.65$, $SD = 1.40$; $t(50) = 5.86$, $p < .001$, $d = .82$, 95% CI [.50, 1.14]. To test the impact of prior confidence on children's learning, we then performed a non-directional independent samples t-test comparing performance in the consistent and inconsistent conditions. Performance did not significantly differ between conditions: consistent confidence $M = 3.75$, $SD = 1.15$; inconsistent confidence $M = 3.55$, $SD = 1.60$; $t(49) = .49$, $p = .625$, *ns*, $d = .14$, 95% CI [-.41, .69].

When combining across both conditions, nondirectional binomial tests (not preregistered) indicate that children significantly favored the currently confident informant on all trials except Trial 4, which was above chance but not significantly so (Trial 1: 86%, $p < .001$; Trial 2: 71%, $p = .005$; Trial 3: 78%, $p < .001$; Trial 4: 63%, $p = .092$, *ns*; Trial 5: 75%, $p = .001$). Results were similar when looking at each condition individually (Consistent condition: Trial 1: 92%, $p < .001$; Trial 2: 67%, $p = .152$, *ns*; Trial 3: 83%, $p = .002$; Trial 4: 58%, $p = .541$, *ns*; Trial 5: 75%, $p = .023$; Inconsistent condition: Trial 1: 81%, $p = .002$; Trial 2: 74%, $p = .019$; Trial 3: 74%, $p = .019$; Trial 4: 67%, $p = .122$, *ns*; Trial 5: 74%, $p = .019$). Observing the pattern of results by trial is especially relevant for the Inconsistent condition, as, if there was a tendency to continue considering past confidence, it would likely have been most evident on the first few trials; yet, participants already overwhelmingly selected the currently confident (and previously hesitant) speaker on Trial 1.

These results indicate that 4- to 5-year-olds preferred to learn from the confident informant in the test phase regardless of their prior confidence. This may suggest that children interpret confidence cues as a situational indicator of knowledge, seeing confidence as a reflection of someone's current knowledge about a specific situation rather than as indicating someone's general knowledge across situations.

## 3. Experiment 2

Results from Experiment 1 suggest that 4- and 5-year-olds attend to confidence cues but use them only (or primarily) to make situational, not enduring, learning decisions. We wished to investigate whether the same pattern would be true of younger children. In Experiment 2, which was ran concurrently with Experiment 1 but with a different age group, we thus replicated Experiment 1 but with children around their third birthday, as this is an age above which they have been shown able to attend to confidence cues and have advanced enough verbal skills to understand our instructions (a pilot study with 13 participants not included in the present sample confirmed that most children of this age could at the very least follow the experiment and remain attentive throughout). However, concerned that the highly verbal scenario presented in Experiment 1 would be too advanced to reveal children's understanding, we added a second part to the experiment involving a non-verbal imitation task that was more similar to those presented to 2- and 3-year-olds in past research [17, 18]. The first part was preregistered prior to data collection (link to preregistration: https://doi.org/10.17605/OSF.IO/HP9EM).

### 3.1 Method

**3.1.1 Participants.** We preregistered a sample size of 64 participants for Part 1. We scheduled more participants than required to account for cancellations and eliminations in both Parts 1 and 2; thus, the final sample consisted of 82 typically-developing 2- and 3-year-olds (31–39 months, $M$ = 35 months, 38 boys and 44 girls), recruited from an in-lab participant database and local daycares between November 14, 2016 and July 4, 2019. Note that demographic information was only collected of in-lab participants. Children were predominately White (seven reported mixed ethnicity, three Asian, one Black and one Indigenous; 25 did not report race/ethnicity) and came in majority from families of average to above-average income (15 did not report family income). The study was either conducted in English (65 children) or in French (17 children). Of the 82 participants, 15 were excluded from Part 1 because of failure to answer one or more trials (12), experimental error (1), or poor language comprehension (2), and 10 were excluded from Part 2 because of camera malfunction (6), experimental error (3), or parental interference (1). Eight additional children were tested but were excluded from both parts for a combination of the reasons above. The final sample sizes were therefore $N$ = 67 and $N$ = 72 for Parts 1 and 2 respectively.

**3.1.2 Materials and procedure.** *Part 1*: *Word learning*. This study was conducted in person and, as in Experiment 1, before participating in the study with their child, parents provided their written consent by completing a consent form. The child's verbal assent was obtained before starting the testing session. The word learning task was also closely modelled on Experiment 1, with a few minor modifications. Specifically, the introduction was shortened to include only two highly familiar pictures in order to encourage children to label objects verbally, and the experimenter placed a picture of each informant on the table to help participants remember the informants' identities and allow them to point to one informant if they were too shy to answer the questions verbally. Additionally, no puppet was presented, and no stickers were mentioned or given.

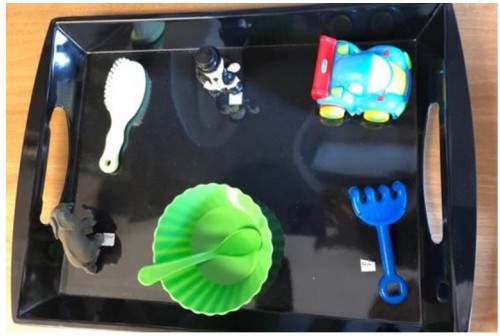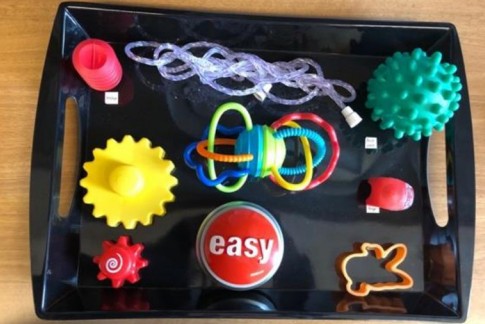

**Fig 1. Experiment 2, Part 2.** Pictures of the trays with familiar (left) and unfamiliar objects (right).

*Part 2*: *Imitation*. For the second part of the study, we created a task that aimed to test the same hypothesis as Part 1 but with reduced verbal demands on young children and a between-subjects design. The task consisted of a Familiar Objects phase and a Novel Objects phase.

In the Familiar Objects phase, the experimenter first showed participants a tray covered by a cloth. The cloth was then removed to reveal six familiar toys (see Fig 1 for pictures of the tray and the six familiar objects). The experimenter then put the cloth back over the tray and introduced an adult male informant on video (hereafter called "A"). Children watched three videos of A playing with three of the familiar toys presented on the tray (the car, the hairbrush, and the bowl/spoon combination). In all three videos, A was portrayed as being either consistently hesitant or confident (between subjects) while playing with the toys. Confidence and hesitance cues were similar to the nonverbal cues used in the first part of the experiment: confidence cues included exclamations "aha!" and "uh-huh!", a raised index finger, a satisfied facial expression, and nodding; hesitance cues included the sounds "hmmm. . ." and "huh?", shrugging of shoulders, and a puzzled facial expression. Afterwards, the experimenter put the tray in front of the participant, removed the cloth, and told the participant they could play with the toys as much as they wanted, as long as they stayed on the table (where a video camera could keep track of the child's actions). The experimenter then started a stopwatch and pretended to be doing work on the laptop to give the child a chance to play freely with the toys. After two minutes had elapsed, children were asked to place the toys back on the tray so that they could take a look at the toys on another tray.

In the Novel Objects phase, the same procedure was repeated but with a tray containing eight unfamiliar toys. The participant first watched videos of A playing with some of the unfamiliar toys (the twisted jump rope, ketchup bottle, gears, and easy button; see Fig 1) again with confidence or with hesitance as in the Familiar Objects phase. Confidence/hesitance during the Familiar and Novel Objects phases were fully crossed, so that approximately a quarter of the children saw A as consistently confident, a quarter consistently hesitant, a quarter confident then hesitant, and a quarter hesitant then confident. Participants were once again given two minutes to play freely with the toys.

## 3.2 Results and discussion

**Part 1: Word learning.** Analyses reported in this section, except in the last paragraph, are preregistered. The main dependent variable was the number of times (out of 5 trials) that children selected the same word as the individual who was confident at test. We also computed, as a secondary dependent variable, the Confidence Difference score; i.e., the difference between performance on the last two trials and the first two trials, in order to see if the confidence at

test has an increasing influence with more test trials. First, we calculated nondirectional independent samples t-tests on both these indices in order to test whether the language of study (English or French) affected performance. Neither was significant (both *p*s>.800).

A directional one-sample *t*-test on the main dependent variable revealed that, contrary to our expectations and to past research, children did not overall prefer to learn from the confident compared to the hesitant individual (*M* = 2.73, *SD* = 1.79; *t*(66) = 1.059, *p* = .147, one-tailed, *ns*). We computed a directional independent samples t-test comparing performance between the consistent and inconsistent conditions, expecting that, if confidence was treated as a person-specific indicator, we would find a greater tendency to trust an individual who was consistently confident over one who is currently confident but was previously hesitant. The mean pattern was actually in the opposite direction: consistent *M* = 2.48, *SD* = 1.75; inconsistent *M* = 2.97, *SD* = 1.82; *t*(65) = -1.11, *p* = .865, one-tailed, *ns*. Finally, it was expected that children in the inconsistent condition would be more likely to switch their trust over to the currently confident individual in the last two test trials compared to the first two test trials, and as a result, the mean Confidence Difference Score would be significantly greater than zero. This was not the case: the mean score was in the opposite direction from what was expected, *M* = -.26, *SD* = .79; *t*(33) = -1.95, *p* = .970, one-tailed, *ns*.

Moreover, although not preregistered, nondirectional trial-by-trial analyses were conducted in order to examine whether there were any other underlying effects. When combining both conditions, participants significantly favored the currently confident informant on the first trial (69%, *p* = .003), but not on any other trial (Trial 2: 55%, *p* = .464; Trial 3: 51%, *p* = 1.000; Trial 4: 46%, *p* = .625; Trial 5: 52%, *p* = .807). When examining each condition separately, similar results were found, though Trial 1 was only significantly above chance for the Inconsistent condition (Consistent condition: Trial 1: 64%, *p* = .163; Trial 2: 48%, *p* = 1.000; Trial 3: 48%, *p* = 1.000; Trial 4: 39%, *p* = .296; Trial 5: 48%, *p* = 1.00, Inconsistent condition: Trial 1: 74%, *p* = .009; Trial 2: 62%, *p* = .229; Trial 3: 53%, *p* = .864; Trial 4: 53%, *p* = .864; Trial 5: 56%, *p* = .608).

**Part 2: Imitation.**   Analyses for Part 2 were not preregistered and should therefore all be considered exploratory. Videos were coded to measure six variables: the number of actions performed by A imitated by the child for familiar (out of 3) and novel objects (out of 4); the amount of time within each two-minute free play phase spent playing with any of the three demonstrated familiar objects or any of the four demonstrated novel objects; and the amount of time spent playing with non-demonstrated familiar and novel objects. For the purpose of coding, any contact with an object was coded as "playing". Note that, because children could spend time playing either with multiple toys at once or not at all, playing times could add up to values greater or smaller than 120 seconds within each phase. Videos from 10 participants were double-coded to verify interrater reliability. Agreement was excellent for the two imitation variables, with 97% agreement for familiar toys and 95% agreement for novel toys. The mean absolute difference between coders in the total playing times with each object category varied between 2.2 and 4.9 seconds.

In the Familiar Objects phase, we first computed a 2x2 mixed ANOVA comparing the amount of time played with toys that were demonstrated or not by the model as a function of the confidence of the model in that phase. We expected that children would prefer to play with the toys used by the model, but that this effect would be moderated by the model's confidence in that phase. We found a significant preference for playing with toys used by the model over the distractor toys, *F*(1,70) = 33.38, *p* < .001. Mean playing time was slightly longer when A was confident than hesitant, but this main effect was not significant, and neither was the interaction effect (see Fig 2). We also compared the number of familiar actions imitated as a

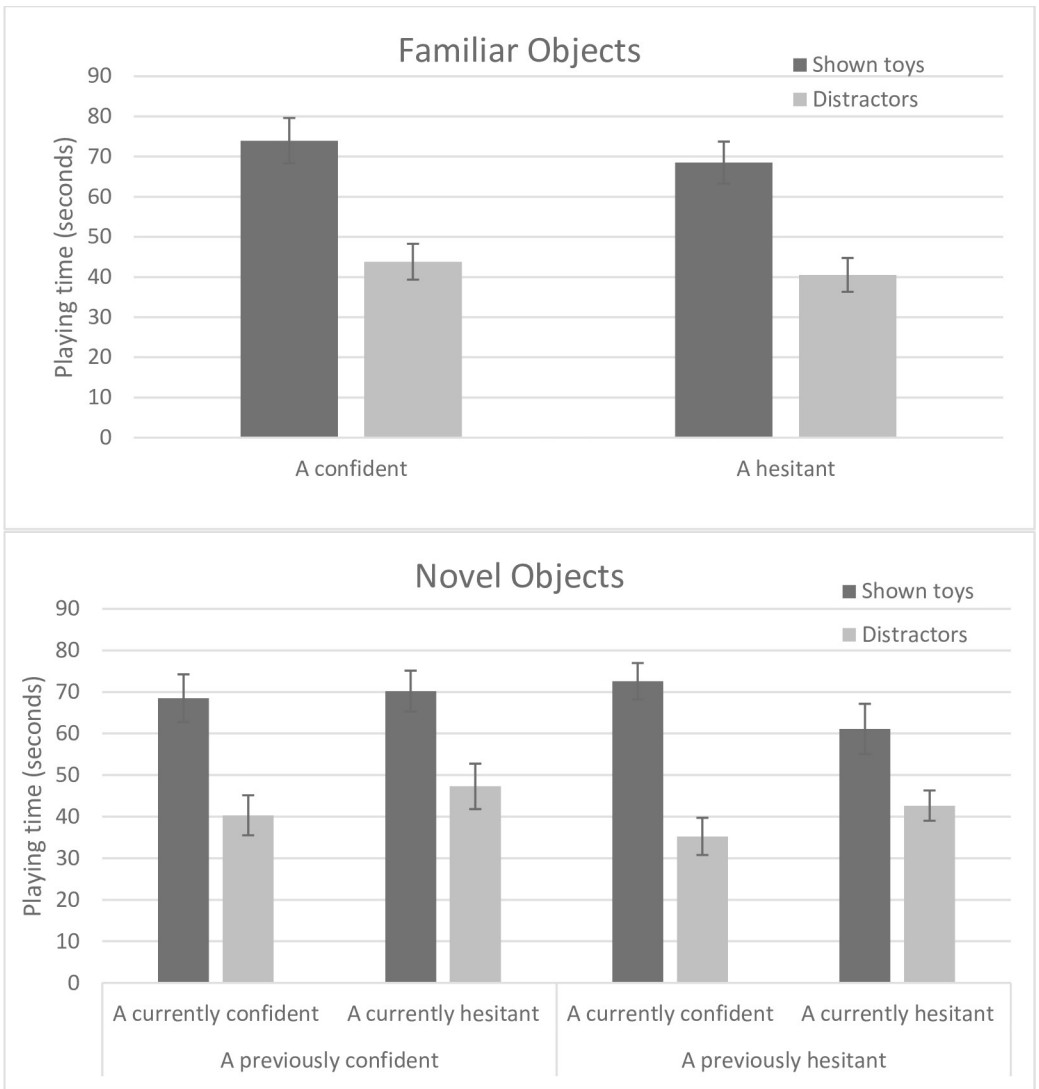

**Fig 2. Experiment 2, Part 2.** Time spent playing with familiar objects (top) and novel objects (bottom) as a function of the type of object (shown toys vs. distractors) and model confidence. Error bars represent standard errors.

function of the model's confidence with an independent-samples t-test: In both conditions, children imitated an average of 1.64 actions out of 3 (*ns*).

In the Novel Objects phase, we computed a 2x2x2 mixed ANOVA on children's playing time with type of toy (demonstrated or not) as a within-subjects variable and current and past confidence as between-subjects variables. Once again, the only significant effect was that children played longer with toys that were demonstrated in videos than with distractors, $F(1,68) = 44.07$, $p < .001$. There were no other significant main or interaction effects (see Fig 2). We also conducted a 2x2 ANOVA on children's imitation of the four novel actions as a function of current and past confidence. The only significant effect, unexpectedly, was a greater tendency to imitate when the model was *previously hesitant* ($M = 2.36$, $SD = 1.42$) than confident ($M = 1.61$, $SD = 1.02$, $F(1,68) = 6.46$, $p = .013$). We do not have a clear explanation for this counterintuitive effect, and, given the number of exploratory analyses presented here, it should be interpreted with caution. Although the means for current confidence were in the predicted

direction (currently confident: $M = 2.16$, $SD = 1.17$; currently hesitant: $M = 1.80$, $SD = 1.39$; $d = .28$, 95% CI [-.19, .75]), this difference was not significant.

These effects seem to suggest that children were influenced by the model's demonstrations, as in both phases they showed more interest in the toys demonstrated by the model than in the distractors. However, playing and imitation did not significantly vary based on the confidence of the model, thus making it unclear whether children were noticing confidence cues at all, let alone formulating any sort of interpretation.

Overall, contrary to previous findings and those obtained with older children, these results suggest that 2- and 3-year-olds did not prefer to learn from the confident informant even when this informant was consistently confident. In fact, in Part 1, participants did not consistently side with either informant (confident or hesitant) except on the very first trial, where children performed in the same direction as those of Experiment 1, i.e., sided with the currently confident informant even when that informant was previously hesitant. It could be that the manipulation was simply too complex for children this young. Therefore, it remains unclear whether younger preschoolers interpret confidence as an indicator of situational or individual knowledge.

## 4. Experiment 3

As younger children did not seem to systematically use the confidence cues presented in our paradigms, we continued focusing on 4- and 5-year-olds only. Results from Experiment 1 suggest that there were some ambiguities even with older children's performance. Indeed, there was a non-significant mean difference between conditions that could be consistent with a small effect of prior confidence continuing to influence older children's performance, but it might have been dwarfed by the influence of the current confidence cues. In other words, it may be that older preschoolers treat confidence *primarily* as situational, but *could* treat it as person-specific if no other more salient cue was present. There is also a possibility that children only relied on current confidence because they did not notice nor remember past confidence; we did not include any memory questions in Experiment 1, which could speak to this possibility. Moreover, the fact that children relied on the current confidence cues to learn novel words in Experiment 1 does not mean that they did not make *any* type of enduring trait attribution. Experiment 3 addresses these possibilities. We devised a different manipulation to examine whether past confidence cues influence older (4-to 5-year-olds) participants' responses when they do not conflict with any other cues. Specifically, at test, participants were read written answers which they were told were written by the informants. Since the answers were single written words that were read in a neutral voice by a third party (the experimenter), no confidence or hesitance cues were present at test. In addition, children were asked explicit attribution and memory questions to better support their interpretation. The word learning phase of this experiment was preregistered prior to data collection; the attribution and memory phases were not preregistered.

### 4.1 Method

**4.1.1 Participants.**   We preregistered a sample size of 40 participants (link to preregistration: https://osf.io/9s63d?view_only=ede9c18920604ac68aa8ad697bcda67c), but as in Experiments 1 and 2, additional children were tested to account for exclusions. Our final sample comprised 41 typically-developing 4- and 5-year-olds (48–71 months, $M_{age} = 59$ months; 21 girls and 20 boys), recruited from an in-lab participant database between December 8, 2019 to March 25, 2022. Children were predominately White (11 reported mixed ethnicity, two Asian, and one Black; four did not report race/ethnicity) and came in majority from families of

average to above-average income (two reported below average income and three did not report family income). The study was either conducted in English (28 children) or in French (13 children). There were four additional children that were excluded from our final sample due to failure to answer one or more trial.

**4.1.2 Materials and procedure.** The current study employed a similar procedure as Experiment 1 but with several important modifications. First, the study began just as the COVID-19 pandemic resulted in a shutdown of in-person research for an extended period; therefore, three participants were tested in-lab and the remaining participants were tested virtually via a videoconferencing app. Second, to broaden our recruitment options, we recruited both English- and French-speaking participants (both languages are widely spoken in the city where the study took place). A French adaptation of the study was thus created (i.e., the script was translated in French and videos of two French-speaking adult female informants were used). Moreover, in Experiment 3, no puppet was presented, and no stickers were mentioned or given, as these were not relevant to our manipulation. Finally, and most importantly for the purpose of the study, the history phase was closely modelled on Experiment 1 but the test phase was modified and extended as described below.

*History phase.* As in Experiment 1, this history phase serves to establish the informants' history of confidence. The same procedures as in Experiment 1 were followed except that there were no introduction videos of the informants; instead, the experimenter introduced each informant via pictures shown on the screen. Participants then watched the same videos as in Experiment 1 of the informants labeling different pictures.

*Test phase–Part 1*: *Word learning.* In each of five *endorse* test trials, a picture of an unfamiliar object was shown to participants. However, rather than seeing the informants label these objects, children were told that the two informants wrote down (conflicting) labels that the experimenter then read to the children. Children were asked for each picture which of the two answers they believed was the correct label.

*Test phase–Part 2*: *Attributions.* For the attributions, children were told the experimenter "had a few questions about B and S." They saw pictures of both and were asked to explicitly choose which one possessed each of six attributes. The first two were knowledge attributes (i.e., "Who knows the names for a lot of bugs?" and "Who knows a lot about stars and planets?"); the next three were about prosocial behaviours (i.e., "Who always says thank you?" and "Who shares her things with her friends and family?"), and affiliations (i.e., "With whom would you like to play?"); the last trial was about a neutral attribution (i.e., "Who doesn't like spaghetti?"), to serve as a distractor.

*Test phase—Memory check.* The last part of the study was a memory check to ensure children were able to recognize the confidence cues and remember them throughout the experiment. Children were asked which of the two informants was previously confident ("Now, remember earlier in the videos, one of them said « Oh, I know! » when she named pictures. Which one said that? B or S?") and which was previously hesitant ("And one of them said "Hmm, I guess?" in the videos. Which one said that? B or S?"). The order of the two memory questions was counterbalanced.

## 4.2 Results and discussion

**Part 1: Word learning.** To determine whether participants relied on the past confidence cues when learning semantic information, we calculated how many times, in 5 trials, children sided with the informant who was confident in the history phase. We then compared the mean to chance (2.5) with a directional one-sample t-test, which revealed that children overall did not prefer to learn from the previously confident informant over the previously hesitant one,

$M = 2.51$, $SD = 1.23$; $t(40) = .06$, $p = .475$, $d = 0.01$, 95% CI [-.30, .32]. Additionally, we conducted trial-by-trial analyses (not preregistered) and results from nondirectional binomial tests reveal that children did not significantly perform above chance on any trials (Trial 1: 56%, $p = .533$; Trial 2: 54%, $p = .755$; Trial 3: 59%, $p = .349$; Trial 4: 37%, $p = .117$; Trial 5: 46%, $p = .755$).

**Part 2: Attributions.** Analyses for the attributions were not preregistered and should therefore all be considered exploratory. Participants did not rely on past confidence cues to learn novel words. To examine whether participants attributed enduring traits to the informants based solely on their past confidence, we conducted binomial tests for each attribution trial. Two participants were not included in these analyses because they did not participate in this phase (although they had completed the word learning phase). Two participants who did not answer the last three attribution questions were only included for the first three questions. Participants did not attribute knowledge (knowledge of bugs: $N$confident = 23/39, $p = .337$; knowledge of stars and planets: $N$confident = 22/39, $p = .522$) nor social traits (saying thank you: $N$confident = 19/39, $p = 1.000$; sharing: $N$confident = 19/37, $p = 1.000$) to either of the previously confident and hesitant informants, but reported that they would prefer to play with the previously confident one ($N$confident = 25/37, $p = .047$). These results suggest that overall, participants performed at chance when making most attributions and thus don't seem to generalize traits that they could have inferred from the past confidence cues to a situation in which no current confidence cues are present.

**Memory check.** To examine whether participants were able to detect and remember past confidence cues, we then performed a binomial test (considering only success or failure at memorizing who was previously confident versus hesitant). We only included children who succeeded at both questions and used 0.5 as chance, although true chance would lead to 0.25, to be conservative and given that children's responses to both questions are unlikely to be completely independent (i.e., pragmatically, if children answered B to the first question, they could feel compelled to answer S to the second question even if they did not remember anything about the History Phase). Results indicate that participants did succeed at the memory check above chance (.5), $N$success = 28 (68%), $p = .014$.

We repeated our analyses for the Word Learning trials looking only at participants who succeeded at the memory check ($N = 28$) to chance. Results revealed that they did not significantly rely on past confidence, $M = 2.79$, $SD = 1.23$; $t(27) = 1.231$, $p = .115$, $d = .24$, 95% CI [-.14, .61]. This suggests that the fact that participants did not rely on past confidence to learn semantic information is not merely because they could not remember the past confidence cues, which seems to support the idea that children perceive confidence cues a situational. Yet, given the direction of the mean, we cannot rule out that there is a small tendency, in children who best remember the History Phase, to treat confidence as person-specific, but that we did not have the statistical power to detect this difference.

For the attribution task, when looking at participants who succeeded at both memory questions, we obtained results that did not seem to significantly differ from chance, though statistical power was admittedly low (knowledge of bugs: $N$confident = 18/27, $p = .122$; knowledge of stars and planets: $N$confident = 19/27, $p = .052$; saying thank you: $N$confident = 11/27, $p = .442$; sharing: $N$confident = 13/26, $p = 1.000$; play with: $N$confident = 18/26, $p = .076$).

Taken together, these results indicate that four- to five-year-olds are able to pay attention and remember confidence cues. However, the findings suggest that, when they are learning novel words and making knowledge and social attributions, they don't systematically favor informants who were more confident in prior situations. These results support the notion that children perceive confidence cues as situational cues to knowledge thus not transferring the inferred knowledgeability from past confidence cues to a current situation without any

indicator of present confidence. Additionally, if confidence was interpreted as person-specific, we would have expected to see knowledge attributions made toward the previously confident informant even when no confidence cues were currently present since it would have been seen as a trait transferrable or generalizable across situation, but it was not the case. Of note, given the results found for one of the social attribution trials (i.e., participants would prefer to play with the previously confident informant), it is possible that children may be able to attribute some specific enduring traits (possibly social traits) to previously confident or hesitant informants. Importantly, this finding may also be a type I error given the usage of a p-value of .05 and 6 tests, and as such, should be interpreted with caution.

We also note that, when only selecting children who correctly answered both memory questions, many results were in the predicted direction for person-specific attributions though not significantly so. We therefore cannot rule out the possibility that there is, in fact, a small tendency to make person-specific attributions that we did not detect because of low power. It is worth noting that correct answers to memory questions can both result from genuinely remembering the History Phase as well as from a coincidental positive bias towards the confident informant (or negative bias towards the hesitant informant)–for instance, some children may just like the look of one informant more than the other. It is thus difficult to unambiguously interpret responses on one type of test trial that select for answers on a different test trial, given that similar irrelevant biases could drive answers on all trials.

## 5. General discussion

In three studies, we investigated whether preschoolers use confidence as a situational or a person-specific knowledge cue. We expected that, were children to use confidence in a strictly situational manner, an individual's *history* of confidence should have no impact on young children's propensity to trust them: only an individual's currently-expressed confidence or hesitance cues should drive children's trust, and if no confidence cues are currently demonstrated children should perform at chance. If, however, confidence was treated as a person-specific knowledge cue, we would expect prior confidence (or lack thereof) to moderate children's use of current confidence cues perhaps dampening their effect when they are in conflict (Experiment 1 and 2) and to guide their current learning process even when no confidence cues are currently present (Experiment 3).

Older preschoolers' results in Experiments 1 and 3 suggest that they treat confidence as a situational indicator in learning situations. Indeed, regardless of individuals' prior confidence or uncertainty, children overwhelmingly preferred to side with whoever was *currently* confident (Experiment 1) and did not favor any informant when the current level of confidence was not available (Experiment 3). Moreover, in Experiment 3, we confirmed that older preschoolers noticed and remembered confidence cues, but they still did not use past confidence at test either in their learning preferences or in their explicit attributions. Given that in Experiment 1 there was a very small mean difference between the Consistent and Inconsistent conditions in the direction expected for a person-specific interpretation, and given that in Experiment 3 those who passed both memory questions did side slightly more often with the previously confident than previously hesitant person, we cannot completely rule out that the prior confidence information had a small impact on children's trust that we simply did not have the power to detect in the present studies. However, even if children do make a person-specific interpretation of confidence at some level, the impact of this interpretation on their learning is minimal compared to that of the situational interpretation.

Additionally, we do not rule out that other types of attributions (other than knowledge attributions) could be made from past confidence cues and that there could then be in some

situations (for instance in social contexts) a person-specific interpretation of confidence cues that is not seen in learning or knowledge-related contexts. It has frequently been assumed by researchers that children pay attention to confidence as a knowledge cue specifically, however, this does not rule out that confidence could *also* be interpreted as a cue to something else, for instance social status, power or emotional trustworthiness. Other studies, perhaps looking more closely at the different types of attributions made towards confident informants could reveal these possible effects.

One may wonder how preschoolers' interpretations as revealed here compare to those of older children or adults. Indeed, it is perhaps not surprising to find preschoolers treating an indicator such as confidence as situational, for several reasons. First, current confidence is likely to be much more salient to a child than any past expressions of confidence or uncertainty simply by being more recent and thus requiring fewer cognitive resources to use. It is possible that one's history of confidence or hesitance would have a stronger or longer-lasting effect on the learning decisions of older individuals who possess greater cognitive resources. Second, preschoolers are known to be less likely than older children to form stable personal trait attributions, at least at an explicit level [44–48]. The tendency to associate one's expressed degree of confidence with the constructs of intelligence or trustworthiness may thus not appear until later in development. Future research could investigate the developmental patterns in the use of cues such as confidence which can conceivably receive multiple interpretations.

It is less obvious what to make of younger children's performance. Contrary to our expectations and to past research, children did not appear to show a preference to learn words or imitate a more confident individual, or, if they did so, it was at a level too weak to be detected given the present study's power. This may suggest that the use of confidence cues to moderate learning is not a strong tendency in such young preschoolers and can easily be disrupted. In fact, exploratory analyses in Part 1, as well as anecdotal evidence from experimenters, suggested that children frequently just repeated the last answer they heard, with the possible exception of the very first trial. This finding is consistent with existing literature suggesting that verbal response to fixed choice questions is related to recency bias in young children and is more likely to be observed as constraints on working memory increase [49, 50]. For instance, Sumner and colleagues [49] found that when 3- and 4-year-olds were given the option to label a toy with one of two novel words ("*X or Y*?"), participants most often chose the last option. Moreover, this recency bias further strengthened as memory demands increased (e.g., increased number of syllables in each novel word). This suggests that the cognitive processing demands (i.e., making a decision involving two novel words) of our task may have been too great for this age group, or at least that the confidence cues were not salient enough to children of this age for them to consider factors beyond the novel words that were presented. While the procedure for Part 2 was perhaps more age-appropriate, it was a new and exploratory methodology and was always administered after Part 1, at a time when children's attentional capacities were perhaps already overwhelmed. Future research can better evaluate the interpretation of confidence in very young children by varying and simplifying the methodology as much as possible.

The present results may be of interest to researchers and practitioners interested in children's learning strategies: In order to, for instance, design educational interventions or make suggestions about how to best influence children's learning, it is not only important to know which learning strategies children can use, but also to know when and why children might use a given strategy. Knowing that preschoolers tend to treat confidence as situational may alleviate concerns that children would treat anyone who has *ever* shown hesitancy as untrustworthy (and should reassure parents, teachers and childcare professionals that sincere expressions of uncertainty when they genuinely do not know the answer to a question should not doom them

to "untrustworthy" status!) However, the present results are not meant to be generalized to all possible learning situations involving confidence cues. Future research should consider applied, naturalistic investigations of children's interpretation of the confidence expressions that they encounter in their real-life conversation partners rather than a single laboratory manipulation featuring strangers.

As all studies, ours are not without limitations. We had just started recruiting for Experiment 3, which aimed to control for some of the limitations of Experiment 1, when the COVID-19 pandemic shut down in-person developmental research. With the prolonged closure, we decided to adapt Experiment 3 for online administration. Because Experiment 3 was conducted online and Experiment 1 in person, we cannot consider both experiments to be fully comparable to one another. However, significant efforts were made to make the experiments as similar as possible, and we did not notice substantially different attrition or any systematic issue that would invalidate either setting. In past research, we have frequently combined data collected at different sites (e.g., in lab versus in daycares with a wide range of set-ups, noise and distraction levels, etc.); the past few years have shown that online research can be conducted with preschool-age children and lead to valid results.

In conclusion, the present studies are to our knowledge the only ones that were specifically designed to test the impact of prior confidence on children's selective trust in an attempt to clarify whether preschoolers treat confidence as situational or person-specific. Our findings suggest that older preschoolers treat confidence cues as situational indicators in learning situations. Failing to perceive confidence as a generalizable indicator of knowledge may contribute to reducing their vulnerability to making misinformed learning decisions. Results of the present studies highlight the need for future research with younger children to better understand the circumstances in which 2- to 3-year-olds attend, or not, to confidence cues.

## Acknowledgments

We would like to thank the following people for their contribution to the research: Sara Amrani, Florence Aquilina, Gladys Ayson, Alexa Burak, Judy Chékiée, Kari-ann Clow, Bianca D'Agostino, Sophie Fobert, Rocksane Forget, Ashley Hewitt, Madison MacLachlan, Marie-Pier Millette, Joana Ntunga Mukunzi, Marie-Lou Ouellette, Suzanne Simba, Michael Slinger, Monica Wharton, Patrice Yazdanyar.

## Author Contributions

**Conceptualization:** Patricia E. Brosseau-Liard.

**Data curation:** Aimie-Lee Juteau, Yasmeen A. Ibrahim, Sara-Emilie McIntee, Rose Varin, Patricia E. Brosseau-Liard.

**Formal analysis:** Aimie-Lee Juteau, Yasmeen A. Ibrahim, Rose Varin, Patricia E. Brosseau-Liard.

**Funding acquisition:** Patricia E. Brosseau-Liard.

**Investigation:** Patricia E. Brosseau-Liard.

**Methodology:** Aimie-Lee Juteau, Patricia E. Brosseau-Liard.

**Project administration:** Aimie-Lee Juteau, Yasmeen A. Ibrahim, Sara-Emilie McIntee, Rose Varin.

**Resources:** Patricia E. Brosseau-Liard.

**Software:** Patricia E. Brosseau-Liard.

**Supervision:** Aimie-Lee Juteau, Patricia E. Brosseau-Liard.

**Validation:** Aimie-Lee Juteau, Patricia E. Brosseau-Liard.

**Visualization:** Patricia E. Brosseau-Liard.

**Writing – original draft:** Aimie-Lee Juteau, Yasmeen A. Ibrahim, Sara-Emilie McIntee.

**Writing – review & editing:** Aimie-Lee Juteau, Yasmeen A. Ibrahim, Sara-Emilie McIntee, Rose Varin, Patricia E. Brosseau-Liard.

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
