## [Decision Letter · Decision Letter 0]

3 Nov 2023

PONE-D-23-23375Do children interpret informants' confidence as person-specific or situational?PLOS ONE

Dear Dr. Juteau,

Thank you for submitting your manuscript to PLOS ONE. After careful consideration, we feel that it has merit but does not fully meet PLOS ONE’s publication criteria as it currently stands. Therefore, we invite you to submit a revised version of the manuscript that addresses the points raised during the review process.

We look forward to receiving your revised manuscript.

Kind regards,

Stefano Triberti, Ph.D.

Academic Editor

PLOS ONE

Journal Requirements:

Did you know that depositing data in a repository is associated with up to a 25% citation advantage (https://doi.org/10.1371/journal.pone.0230416)? If you’ve not already done so, consider depositing your raw data in a repository to ensure your work is read, appreciated and cited by the largest possible audience. You’ll also earn an Accessible Data icon on your published paper if you deposit your data in any participating repository (https://plos.org/open-science/open-data/#accessible-data).

**Additional Editor Comments:**

The article received quite positive comments from the Reviewers and I agree it is worthy of consideration. I encourage Authors to take into account Reviewers' corrections and suggestions for revision. 

Reviewers' comments:

Reviewer's Responses to Questions

**Comments to the Author**

1. Is the manuscript technically sound, and do the data support the conclusions?

Reviewer #1: Yes

Reviewer #2: Yes

2. Has the statistical analysis been performed appropriately and rigorously? 

Reviewer #1: Yes

Reviewer #2: Yes

3. Have the authors made all data underlying the findings in their manuscript fully available?

Reviewer #1: Yes

Reviewer #2: Yes

4. Is the manuscript presented in an intelligible fashion and written in standard English?

Reviewer #1: Yes

Reviewer #2: Yes

5. Review Comments to the Author

Reviewer #1: This paper reports the results of three studies examining whether children treat confidence as a situational or person-specific cue to knowledge/credibility (in guiding children's selective social learning). The paper is very well written; the methods are clearly presented and results reported clearly and concisely. I think the research question is an important one in furthering our understanding of how/when children use confidence as a cue to credibility, and the results presented here will make a useful contribution to the literature on this subject.

Really, I think this paper is essentially ready for publication. My one comment is that the analyses, while they are sound as presented, could be streamlined a bit and made more parsimonious. That is, instead of the repeated variants of t-tests (or ANOVAs in the later experiment) that tackle the same underlying data in slightly different ways - the authors could opt for simply analysing the trial by trial data using a mixed effect logistic regression (this would allow for analysis of the overall tendencies and specific trial by trial data in a single model). I don't see this is a necessary change, but it could streamline the paper somewhat (if that's something the editorial team would like in this instance).

My only suggestion for the authors is to the include confidence intervals around all estimated effect sizes - those, in line with the release of the data on the OSF, will help for future researchers wanting to build on this research (or include it in a meta-analysis).

All in all, well written, clear, and informative paper. Great work!

Reviewer #2: The paper “Do Children Interpret Informant Trust as Person-Specific or Situational?” presents itself as interesting experimental research aimed at investigating the interpretation of confidence signals by 4 and 5 year old children. In particular, the aim of the study is to verify whether children of this age perceive trust signals as situational or as person specific. The results show that 4–5-year-olds treat trust signals primarily as situational. The manuscript is generally well written.

I think the article could be considered for publication after minor revisions; In particular:

- Introductory section:

Line 46: I suggest better specifying the age of the samples of children investigated in the studies cited.

Line 131: I suggest delving deeper into the explanation of Moore and colleagues' research.

- Methodology

I suggest clarifying the methodology used (the three conditions and the different phases) through a table or a small graphic.

- Discussion

To justify the results in parts different from those emerged in previous studies, I suggest adding studies that investigate the relationship between study results and memory development in childhood. Finally, I suggest including a section in which to propose how these results could be useful for the purposes of structuring educational programs and technologies for the age group considered.

6. PLOS authors have the option to publish the peer review history of their article (what does this mean?). If published, this will include your full peer review and any attached files.

Reviewer #1: No

Reviewer #2: No

---

## [Author Response · Author response to Decision Letter 0]

11 Jan 2024

Editor’s comments

Response: The style requirements have been reviewed.

Response: We have deposited our minimal data set underlying the results of our manuscript in a public repository (available here: https://osf.io/9a4j2/?view_only=dc7e670c756c41ceb8e84735faf397ad). It is to be noted that upon verifying that all data supporting our findings were available, we rectified the number of participants included in Experiment 2 which slightly changed results (e.g., means, standard deviations) but does not impact our conclusions.

3. In your Data Availability statement, you have not specified where the minimal data set underlying the results described in your manuscript can be found.

Response: Our minimal data set will be available on the Open Science Framework upon acceptance (private link: https://osf.io/9a4j2/?view_only=dc7e670c756c41ceb8e84735faf397ad). We would thus like to change our Data Availability as followed: The data that support the findings of this study is available in a public repository (Open Science Framework) and can be found here [link available at acceptance].

Response: We appreciate this opportunity to change our Data Availability statement and as previously indicated, we would thus like to change it as followed: The data that support the findings of this study is available in a public repository (Open Science Framework) and can be found here [link available at acceptance].

Response: The reference list has been reviewed and is complete and correct. Because we had cited a paper that had been submitted but has not yet been accepted, we removed this reference (see line 610). In response to one of Reviewer #2’s suggestions, we also added 2 references (see line 639). As a result, our numbered in-text citations and references were rearranged.

Reviewer #1 

1. My one comment is that the analyses, while they are sound as presented, could be streamlined a bit and made more parsimonious. That is, instead of the repeated variants of t-tests (or ANOVAs in the later experiment) that tackle the same underlying data in slightly different ways - the authors could opt for simply analysing the trial by trial data using a mixed effect logistic regression (this would allow for analysis of the overall tendencies and specific trial by trial data in a single model). I don't see this is a necessary change, but it could streamline the paper somewhat (if that's something the editorial team would like in this instance).

Response: We agree with Reviewer #1 that logistic mixed-effects models would be a desirable strategy with the present data, however we chose to keep our analyses unchanged in order to remain consistent with our pre-registrations.

2. My only suggestion for the authors is to the include confidence intervals around all estimated effect sizes - those, in line with the release of the data on the OSF, will help for future researchers wanting to build on this research (or include it in a meta-analysis).

Response: We have added confidence intervals around all estimated effect sizes.

Reviewer #2 

1. Introductory section:

Line 46: I suggest better specifying the age of the samples of children investigated in the studies cited.

Response: We specified the age of the samples from the studies cited (e.g., see lines 49, 65 and 69).

2. Line 131: I suggest delving deeper into the explanation of Moore and colleagues' research.

Response: We provided additional information regarding Moore and colleagues’ research (see lines 134-143).

3. Methodology

I suggest clarifying the methodology used (the three conditions and the different phases) through a table or a small graphic.

Response: We clarified our methodology by adding a table (Table 1) summarizing the 

three conditions and the different phases.

4. Discussion

To justify the results in parts different from those emerged in previous studies, I suggest adding studies that investigate the relationship between study results and memory development in childhood.

Response: We added information in the General Discussion on the potential relationship between our results and memory development in childhood (see page 28). 

5. Finally, I suggest including a section in which to propose how these results could be useful for the purposes of structuring educational programs and technologies for the age group considered.

Response: We included such a section in the revised General Discussion (see page 29).

---

## [Editor Report · Decision Letter 1]

22 Jan 2024

Do children interpret informants' confidence as person-specific or situational?

PONE-D-23-23375R1

Dear Dr. Juteau,

We’re pleased to inform you that your manuscript has been judged scientifically suitable for publication and will be formally accepted for publication once it meets all outstanding technical requirements.

Kind regards,

Stefano Triberti, Ph.D.

Academic Editor

PLOS ONE